Prediction of the mechanism of miRNAs in laryngeal squamous cell carcinoma based on the miRNA-mRNA regulatory network

Ma Jinhua
Hu Xiaodong
Dai Baoqiang
Wang Qiang
Wang Hongqin hongqinwang0317@126.com
Department of Otolaryngology, Cangzhou Central Hospital , Cangzhou , China
Nohata Nijiro
Electronic publication date: 2021 Aug 24
Publication date: 2021
Volume: 9
Electronic Location ID: e12075
Received 2021 Mar 31; Accepted 2021 Aug 6
Copyright: ©2021 Ma et al.
Copyright year: 2021
Copyright holder: Ma et al.
License: This is an open access article distributed under the terms of the Creative Commons Attribution License, which permits unrestricted use, distribution, reproduction and adaptation in any medium and for any purpose provided that it is properly attributed. For attribution, the original author(s), title, publication source (PeerJ) and either DOI or URL of the article must be cited.
License URL: https://creativecommons.org/licenses/by/4.0/

Keywords: Laryngeal squamous cell carcinoma, Bioinformatics analysis, miRNA-mRNA regulatory network, miR-140-3p

Funding: The Hebei Medical Research Youth Program 20200175 This work was supported by the Hebei Medical Research Youth Program (20200175). The funders had no role in study design, data collection and analysis, decision to publish, or preparation of the manuscript.

==============================
In this study, a bioinformatics analysis is conducted to screen differentially expressed miRNAs and mRNAs in laryngeal squamous cell carcinoma (LSCC). Based on this information, we explored the possible roles of miRNAs in the pathogenesis of LSCC. The RNA-Seq data from 79 laryngeal cancer samples in the Gene Expression Omnibus (GEO) database were sorted. Differentially expressed miRNAs and mRNAs in LSCC are screened using the PERL programming language, and it was analysed by Gene Ontology (GO) and the Kyoto Encyclopedia of Genes and Genomes (KEGG). The miRNA-mRNA regulatory network of LSCC is constructed using Cytoscape software. Then, quantitative real-time PCR (QRT- PCR), Cell Counting Kit-8 (CCK8) and flow cytometry analysis we are used to further validate key miRNAs. We identified 99 differentially expressed miRNAs and 2,758 differentially expressed mRNAs in LSCC tissues from the GEO database. Four more important miRNAs displaying a high degree of connectivity are selected, these results suggest that they play an important role in the pathogenesis of LSCC. As shown in the present study, we identified specific miRNA-mRNA networks associated with the occurrence and development of LSCC through bioinformatics analysis. We found a miRNA molecule closely related to LSCC based on miRNA-mRNA network: miR-140-3p was down-regulated in LSCC. In addition, the potential antitumor effect of miR-140-3p in LSCC was verified in the experiment, and it was proved that overexpression of miR-140-3p could inhibit the proliferation of LSCC cells and promote cell apoptosis, suggesting that miR-140-3p may be a potential tumor marker in LSCC.

Introduction

Head and neck squamous cell carcinoma (HNSCC) is the sixth most common malignancy, and LSCC is a common malignant tumor of HNSCC deriving from the laryngeal mucosal epithelium (Lampri et al., 2015; Wei et al., 2018; Xiong et al., 2020). LSCC causes great pain in patients, seriously threatens the health of humans, and increases the economic burden on patients and the society. In 2015, there were approximately 25,300 cases of laryngeal cancer in China; and the incidence rate of laryngeal cancer ranked the 21st in all malignant tumors, most of these patients are middle-aged and elderly men (He et al., 2020). At present, the primary diagnosis of LSCC mainly depends on the medical history, physical examination of the head and neck, laryngoscopy, imaging examination and other auxiliary examinations, and the diagnosis of LSCC must also rely on the pathological tissue biopsy results. Previous literature also showed that miR-941, miR-21 and miR-27a were potential diagnostic markers of LSCC (Yu et al., 2014; Zhao et al., 2020). Surgery is still the main treatment for LSCC, and the survival rate of LSCC has not improved significantly in recent years, which is caused by the lack of understanding of the mechanism of occurrence and progression of LSCC (Chen et al., 2016). With the development of research, more and more microRNAs (miRNAs) have opened a mysterious door for the diagnosis and treatment of LSCC. Therefore, it is imperative to further explore and elucidate the pathogenesis of LSCC, identify new biomarkers, and study new effective therapeutic targets.

MiRNAs are an endogenous single-stranded noncoding RNA that regulate gene expression, with a length of approximately 18 to 22 nucleotides. These non-coding RNAs were discovered by Lee et al. in 1993 (Lee, Feinbaum & Ambros, 1993). In the transcription process, the miRNA gene is first transcribed in the nucleus to form the original transcript, namely, a pri-miRNA with stem ring structure of approximately 300 to 1000 nucleotides. Subsequently, the original transcript is processed through further splicing into a functional, mature miRNA and transported into the cytoplasm, where it selectively binds to complementary mRNAs to inhibit protein production and regulate gene expression. Notably, miRNAs are highly conserved and widely exist in animals, plants, viruses and other organisms (Baril, Ezzine & Pichon, 2015; Vishnoi & Rani, 2017). Studies have shown that miRNAs regulate approximately one-third of the mRNAs in the body and thus participate in a variety of biological processes in humans (Wang et al., 2020). A large number of studies have shown that miRNAs play an important role in inhibiting or promoting cancer growth by regulating the mRNAs encoded by tumour suppressor genes or oncogenes, affecting the occurrence, development, metastasis and recurrence of tumours (Wu et al., 2016; Qadir & Faheem, 2017; Wang et al., 2020). Notably, miRNAs influence the occurrence, development and metastasis of LSCC through complex mechanisms, and thus the discovery of new miRNAs in LSCC tissues is very important for the diagnosis and treatment of LSCC (Liu & Ye, 2019). The purposes of this study were to screen for differentially expressed miRNAs and mRNAs in LSCC using molecular biology and bioinformatics techniques, to successfully predict miRNA target genes, and to construct a miRNA-mRNA regulatory network based on the relationships between miRNAs, mRNA, of miRNA target genes and differential expression (Guan et al., 2015; Fei et al., 2017; Ma et al., 2021). Next, further evaluation was performed using gene ontology (GO) and Kyoto Encyclopedia for gene and genome (KEGG) path analysis. Then, we performed QRT-PCR, CCK8 and flow cytometry analysis to further validate key miRNAs. This study provides a new method to explore the pathogenesis of LSCC at the molecular level, as well as a new reference and direction for the search for molecular markers to diagnose LSCC and new therapeutic methods.

Material and Methods

Sample source

LSCC miRNA and mRNA chip data were collected from the GSE124678 and GSE59102 retrieved from the National Center for Biotechnology Information (NCBI) GEO Datasets (http://www.ncbi.nlm.nih.gov/geo). Derived from the GPL16770 platform, GSE124678 includes five normal tissue samples and 32 tumor tissue samples. Derived from the GPL6480 platform, GSE59102 includes 13 normal tissue samples and 29 tumor tissue samples. The miRNA and mRNA microarray data of LSCC and normal laryngeal tissues were processed using platform annotation file (Edgar, Domrachev & Lash, 2002). We then used PERL5.30.2 (https://www.PERL.org/) to analyze and process the dataset; according to the annotation platform file of the expression profile, the probe name was converted to the corresponding gene name, and the empty probe was removed. Then, the miRNA and mRNA data were obtained by sequencing the data from the two datasets in order of the normal group and the tumor group.

Analysis of differentially expressed miRNAs and mRNAs

We used the LIMMA software package from R software (https://rstudio.com/, ver. 3.6.2). Objective to screen miRNAs and mRNAs differentially expressed in LSCC (Colaprico et al., 2016) The screening criteria we set are: —log2(fold change)—>1 and FDR (False Discovery Rate)<0.05. Then, the “pheatmap” package and “ggplot2” package in R (3.6.2) were used to produce the volcano map. The differentially expressed miRNAs and mRNAs were screened.

Prediction of miRNA targets

The effects of miRNAs are mediated by their complete or incomplete interactions with target genes, affect the expression of target genes. Hence, miRNAs target genes must be predicted. In this study, 99 differentially expressed miRNAs target genes were predicted by FunRich (3.1.3).

Construction of the miRNA-mRNA regulatory network diagram

First, we used PERL software to intersect the target genes of miRNAs with the differentially expressed mRNAs in LSCC to obtain the common mRNAs and corresponding miRNAs. Previously, a negative regulatory relationship between miRNAs and their target genes was identified (Bartel, 2009). Therefore, based on the negative regulatory relationship between miRNAs and mRNAs, miRNA-mRNA pairs with significant differences in log2 (fold change) values were selected, and Cytoscape (3.7.2) software was used to visualize the regulatory network between the two to obtain the miRNA-mRNA regulatory network (Lin & Chen, 2018; Zhao et al., 2020).

GO and KEGG enrichment analyses

We performed GO and KEGG enrichment analyses of differentially expressed mRNAs to further clarify the biological functions of target genes in the regulatory network (Kanehisa & Goto, 2000; Khatri, Sirota & Butte, 2012). R software in the Bioconductor plug-in (http://www.bioconductor.org) was used to perform the GO and KEGG enrichment analyses with the criteria of p < 0.05 and q < 0.05. The results are given in the form of bar chart.

Cell culture and cell transfection

We used Human LSCC cell line LSC-1 (Bluefbio, China) and human laryngeal epithelial cells (HLEC; Lifeline, America). Dulbecco’s Modified Eagle’s medium (DMEM) was used as the basic medium and 10% foetal bovine serum (FBS) and 100 g/ml penicillin/streptomycin were added to the medium, the culture conditions were 37 °C and 5% CO2. In this study, the cells were all used at passages 2–4 after recovery.

A total of 5 nmol miR-140-3p miRNA mimics (miR-140-3p) or miRNA mimic negative control (miR-NC) (GENECREATE, Wuhan, China) was dissolved in 0.9%NaCl as a storage solution of 20 µM. LSC-1 cells (3*105 cells per well) were cultured in 6-well plates overnight. When cell density reached 60–70%, the cell medium was replaced with a complete medium without antibiotics. 1.5 µL of miR-140-3p or miR-NC storage solution was diluted with 50 µL serum-free medium, and 1 µL Lipofectamine2000 (Beyotime, Shanghai, China) was added to the other 50 µL serum-free medium. Mix well and let stand at 37 °C for 5 min. Mix the two tubes together and let stand at 37 °C for 20 min. Then add the mixture into each cell well, gently shake the culture plate to mix the complex with the cells, add 500 µL complete medium, and place in the cell incubator for further culture for 48 h, then collect the cells for subsequent experiments. All cells were cultured in complete medium for at least 24 h before transfection and rinsed with phosphate-buffered saline (PBS, pH 7.4) before transient transfection. The generated cell clones were tested for stable miR-140-3p expression.

RNA extraction and QRT- PCR

According to the manufacturer’s instructions, total RNA was extracted from the cell using Trizol reagent (TAKARA, Japan). Total RNA was reverse transcribed into cDNA using the PrimeScript RT Reagent Kit (Takara, Japan). Reverse transcription: 25 °C10 min, 50 °C for 30 min and 85 °C for 5 min. Expression was detected using the fluorescence quantitative PCR kit and the following conditions: there are 40 cycles at 95 °C for 5 min, 95 °C for 10 s, 60 °C for 30 s. The solubility curve temperature was set to 60−95 °C, and three replicate wells were set for each specimen. The expression of miR-140 was normalized to a small nuclear RNA (U6) as an internal reference. The results were calculated using the 2−ΔΔCt method. The primer sequences used for quantitative real-time PCR analyses of miR-140-3p were as follows: forward, 5′-ACACTCCAGCTGGGTACCACAGGGTAGAA-3′ and reverse, 5′- CTCAACTGGTGTCGTGGAGTCGGCAATTCAGTTGAGCCGTGGTT-3′. The primer sequences for small nuclear RNA (u6) are as follows: forward, 5′- CTCGCTTCGGCAGCACA -3′ and reverse 5′- AACGCTTCACGAATTTGCGT -3′. The relative expression of miRNA was detected by 2−ΔΔCt.

CCK8 assay

We performed the CCK8 assay to measure the proliferation of LSC-1 cells. Cells were cultured in 96-well culture plate, and the inoculation density was 3*105/well (Jiang et al., 2019). For cell transfection, cells were cultured overnight. After 48 h of transfection, 10 mL of CCK8 solution was added to each well, and the cells were incubated at 37 °C for another 60 min. The absorbance of the solution was measured at 490nm by Smart Microplate Reader (SMR) 16.1.

Flow cytometry analysis

LSC-1 cells were transfected for 48 h. After transfection with miR-140-3p mimics or control miRNA, LSC-1 cells were washed with cold PBS buffer. Then, LSC-1 cells were detached with trypsin and washed twice with cold PBS. According to the manufacturer’s instructions, cells were subsequently double stained with FITC-labelled Annexin V and propidium iodide (PI) using the FITC Annexin V Apoptosis Detection Kit (BD Biosciences, USA). Flow cytometer was performed to determine the percentage of apoptotic cells (FACS Calibur; BD Biosciences, USA). The data were analyzed by FLOWJ software.

Statistical analysis

GraphPad Prism5.0 and statistical product and service solutions (SPSS) 22.0 software were used for statistical analyses. Use t-test to enumeration data, and P < 0.05 indicated the significance level.

Results

Differentially expressed miRNAs and mRNAs

After annotating the GSE124678 dataset, 1205 pieces of miRNA information were obtained from human laryngeal carcinoma tissues, including 5 tissues in the normal group and 32 tissues in the tumour group. 99 differentially expressed miRNAs were screened by PERL language (Table S1 for details). A volcanic map of differentially expressed miRNAs is then drawn using R software (Fig. 1). Similarly, we used the GSE59102 dataset from the LSCC mRNA chip downloaded from the GEO database to create a volcanic map of the differentially expressed mRNAs for human LSCC (Fig. 2). Thirteen tissues in the normal group and 29 tissues in the tumour group were analysed. A total of 2758 differentially expressed mRNA, were obtained, including 1312 up-regulated and 1446 down-regulated (Table S2 for details).

Figure 1 Volcanic map of differentially expressed miRNA.

Orange represents upregulated miRNAs, blue represents downregulated miRNAs, and black represents miRNAs without a significant difference in expression.

Figure 2 Volcanic map of differentially expressed mRNAs.

Orange represents upregulated mRNAs, blue represents downregulated mRNAs, and black represents mRNAs without a significant difference in expression.

Figure 3 The miRNA-mRNA regulatory network.

Ellipses represent mRNAs, triangles represent miRNAs, orange represents upregulated expression, blue represents downregulated expression, and connected lines represent targeted relationship.

Prediction of miRNA targets

FunRich (3.1.3), a gene function analysis tool, was used to predict the target genes of the 99 differentially expressed miRNAs; 1386 target genes were obtained in the background, and the number of miRNAs matched with them was 32.

Construction of the miRNA-mRNA network diagram

The differentially expressed mRNAs and predicted miRNA target genes in laryngeal carcinoma were processed using PERL. The intersection of miRNA target gene and mRNAs, that is, the targeted regulation relationship between miRNAs and mRNAs, was obtained. Then, Cytoscape software was used to map the miRNA-mRNA regulatory network (Fig. 3). This miRNA-mRNA regulatory network contains 10 miRNAs and 96 mRNAs (Table S3 for details). The more genes that are connected with miRNAs in this regulatory network indicate that this miRNA is very important in the occurrence and development of LSCC. As this regulatory network clearly shows, miR-140-3p regulates more mRNAs. It is suggested that miR-140-3p plays an important role in the occurrence and development of LSCC.

Functional enrichment analysis

We conducted GO and KEGG enrichment analyses of the 2758 differentially expressed mRNAs using p < 0.05 and q < 0.05 as screening conditions. The GO analysis includes three categories: molecular function (MF), cellular component (CC), and biological process (BP). These differentially expressed mRNAs promoted the occurrence and development of LSCC by participating in various BPs, CCs and MFs (Fig. 4). Similarly, the results of KEGG enrichment analysis were also included Cell cycle (hsa04110), p53 signaling pathway (hsa04115), IL-17 signaling pathway (hsa04657), chemical carcinogenicity (hsa05204), etc (Fig. 5, Table 1).

Figure 4 GO enrichment analysis of differentially expressed mRNAs.

Figure 5 KEGG enrichment analysis of differentially expressed mRNAs.

Table 1 GO and KEGG pathway enrichment analysis of differentially expressed mRNAs in LSCC.

Pathway ID	Pathway description	P-Value	Count	
GO:0043062	extracellular structure organization	5.33E−19	114	
GO:0030198	extracellular matrix organization	5.81E−19	104	
GO:0140014	mitotic nuclear division	3.60E−14	75	
GO:0000280	nuclear division	1.08E−12	97	
GO:0048285	organelle fission	6.24E−12	102	
GO:0001503	ossification	6.03E−11	91	
GO:0033260	nuclear DNA replication	4.25E−10	26	
GO:0062023	collagen-containing extracellular matrix	6.46E−25	122	
GO:0044420	extracellular matrix component	3.96E−16	30	
GO:0005604	basement membrane	3.44E−12	37	
GO:0000775	chromosome, centromeric region	1.75E−11	56	
GO:0016323	basolateral plasma membrane	2.77E−10	58	
GO:0000779	condensed chromosome, centromeric region	3.07E−10	39	
GO:0000776	kinetochore	5.46E−10	42	
GO:0005201	extracellular matrix structural constituent	1.52E−17	62	
GO:0048018	receptor ligand activity	4.03E−11	109	
GO:0005125	cytokine activity	6.52E−10	60	
GO:0008201	heparin binding	8.39E−10	50	
GO:0005539	glycosaminoglycan binding	1.29E−09	61	
GO:0030414	peptidase inhibitor activity	1.01E−07	48	
GO:0004857	enzyme inhibitor activity	1.09E−07	81	
hsa04060	Cytokine-cytokine receptor interaction	4.96E−10	77	
hsa03030	DNA replication	8.60E−08	18	
hsa04110	Cell cycle	1.75E−07	38	
hsa04512	ECM-receptor interaction	9.49E−07	29	
hsa05222	Small cell lung cancer	2.65E−06	29	
hsa04061	Viral protein interaction with cytokine and cytokine receptor	5.56E−06	30	
hsa00830	Retinol metabolism	7.66E−06	23	
hsa05146	Amoebiasis	8.64E−06	30	
hsa00982	Drug metabolism - cytochrome P450	2.22E−05	23	
hsa00980	Metabolism of xenobiotics by cytochrome P450	2.99E−05	24	
hsa05204	Chemical carcinogenesis	3.07E−05	25	
hsa04657	IL-17 signaling pathway	3.84E−05	27	
hsa04970	Salivary secretion	8.78E−05	26	
hsa04974	Protein digestion and absorption	0.0002	27	
hsa04914	Progesterone-mediated oocyte maturation	0.0003	26	
hsa04115	p53 signaling pathway	0.0007	20	
hsa04610	Complement and coagulation cascades	0.0009	22	
hsa05150	Staphylococcus aureus infection	0.0009	24	
hsa02010	ABC transporters	0.0011	14	
hsa04933	AGE-RAGE signaling pathway in diabetic complications	0.0017	24	
hsa04510	Focal adhesion	0.0018	41	
hsa04062	Chemokine signaling pathway	0.0026	39	

The expression of miR-140-3p in LSC-1 cells is low

In this study, the expression of miR-140-3p in LSC-1 cells and HLECs was detected using QRT- PCR. We examined the expression of miR-140-3p in HLECs and LSC-1 cells, and observed lower miR-140-3p expression in LSC-1 cells than in HLECs. Obviously, these experimental results suggest that the decrease in miR-140-3p expression in LSC-1 cells. The results were consistent with bioinformatics analysis (Fig. 6).

Upregulation of miR-140-3p expression inhibits LSC-1 cells proliferation in vitro

The CCK8 assay was performed to further assess the biological role of miR-140-3p in LSC-1 cells. LSC-1 cells were transfected with miR-NC or miR-140-3p mimics. Cell proliferation was detected by CCK8 method. QRT- PCR results showed that the transfection of miR-140-3p mimics led to overexpression of miR-140-3p in LSC-1 cells compared with the control group (Fig. 7). Similarly, the results of the CCK8 assay showed significantly impaired growth of LSC-1 cells transfected with miR-140-3p mimics compared with LSC-1 cells transfected with miR-NC. Based on these results, the high expression of miR-140-3p in LSC-1 cells inhibited their proliferation (Fig. 8).

The upregulation of miR-140-3p promoted the apoptosis of LSC-1 cells

Flow cytometry was used to further clarify the effect of miR-140-3p upregulation on the apoptosis of LSC-1 cells. Compared with miR-NC, miR-140-3p mimics significantly promoted the apoptosis of LSCC cells. Thus, the upregulation of miR-140-3p promoted the apoptosis of LSC-1 cells (Fig. 9).

Discussion

In recent years, the incidence of LSCC and other types of malignant tumours has been increasing year by year, while the survival rates of patients with LSCC have improved with advances in medical technology for LSCC, such as surgery, radiotherapy and chemotherapy and targeted therapy; however, the complications after surgery, radiation and chemotherapy still afflict patients with LSCC (Jin et al., 2011). Therefore, understanding LSCC from a molecular perspective is particularly important (Huang et al., 2020).

Importantly, miRNAs regulate approximately one-third of human genes and play a key role in the pathogenesis of malignant tumours. According to numerous studies, miRNAs are abnormally expressed in the tissues and cells of malignant tumours, examples include breast cancer (Wang et al., 2019), gastric cancer (Chen et al., 2019), thyroid cancer (Li et al., 2013), colorectal cancer(Balacescu et al., 2018) and cervical cancer (Qu et al., 2018). Based on accumulating evidence, the expression of miRNA is closely related to the prognosis of laryngeal carcinoma (Li et al., 2016a; Li et al., 2016b; Zhang, Fu & Zhang, 2018). Moreover, miRNAs such as miR-199b-5p, miR-424-5p, miR-1297 and miR-145-5p have been found to further participate in the occurrence and progression of LSCC by regulating their respective target genes (Gao et al., 2019; Li et al., 2019; Ashirbekov et al., 2020). Thus, miRNAs may be new biomarkers for the occurrence and progression of LSCC.

Figure 6 The expression of miR-140-3p in HLECs and LSC-1 cells was measured using the QRT- PCR assay.

(* p < 0.05, ** p < 0.01, and *** p < 0.001).

Figure 7 MiR-140-3p and miR-NC were transfected into LSC-1 cells, respectively, and the expression of miR-140-3p was detected by QRT-PCR.

(* p < 0.05, ** p < 0.01, and *** p < 0.001).

Figure 8 Effects of miR-140-3p on LSC-1 cell proliferation in vitro.

* p < 0.05.

Figure 9 LSC-1 cells were transfected with miR-140-3p and miR-NC, respectively, and Annexin V-FITC and PI staining were performed to detect the percentage of cell apoptosis by flow cytometry.

(* p < 0.05, ** p < 0.01, and *** p < 0.001).

In this study, bioinformatics was used to analyse two GEO datasets, and 99 miRNAs and 2758 mRNAs were screened for differential expression between the normal group and the tumour group. Then, the regulatory networks of miRNAs and mRNAs in LSCC were systematically analysed. Functional GO and KEGG enrichment analyses revealed the potential roles of non-coding RNAs and coding RNAs in the development of LSCC. On the basis of the negative regulatory relationship between miRNAs and mRNAs, we constructed a network diagram containing 10 miRNAs and 96 mRNAs for the purpose of better understanding the pathogenesis of LSCC. From the network diagram, we conclude that miR-140-3p is the miRNA that is most densely connected to other mRNAs in the targeted regulatory network.

Initially, Wienholds et al. (2005) identified a role for miR-140 in cartilage development in vivo. Notably, miR-140-3p belongs to the miR-140 cluster and has been shown to play an important role in the occurrence and development of a variety of tumours; miR-140-3p inhibits the proliferation of human cervical cancer cells by targeting RRM2 to induce cell cycle arrest and early apoptosis (Ma, Zhang & Sun, 2020). Upregulation of miR-140 inhibits the proliferation and invasion of colorectal cancer (Zhang et al., 2015), and miR-140-3p expression is decreased in patients with breast cancer (Salem et al., 2016). Additionally, miR-140-3p inhibits the growth of colorectal cancer cells and promote apoptosis by regulating programmed cell-death 1 ligand 1 (PD-L1) (Jiang et al., 2019). Nevertheless, the expression and function of miR-140-3p in LSCC development remains unclear. In this study, we found that the expression of miR-140-3p decreased in LSCC. In addition, overexpression of miR-140-3p significantly reduced proliferation and induced LSC-1 cells apoptosis in vitro. Based on these results, miR-140-3p plays an important role in the occurrence and progression of LSCC. Therefore, we propose that miR-140-3p plays a potentially important role in the development of LSCC cells.

Finally, the results obtained from the KEGG enrichment analysis again verified the results of the GO analysis. In the GO term enrichment analysis, the differentially expressed mRNAs were importantly connected with the terms ‘extracellular matrix organization’, ‘extracellular matrix component’, ‘nuclear DNA replication’ and ‘mitotic nuclear division’. KEGG pathway analysis indicated that the roles of the differentially expressed mRNAs were enriched in ‘ECM-receptor interaction’, ‘DNA replication’, ‘Cell cycle’, ‘p53 signalling pathway’ and ‘complement and coagulation cascades’. The extracellular matrix (ECM) is composed of and interlocking mesh of water, minerals, proteins secreted by resident cells, which is responsible for cell–cell communication, cell adhesion and cell proliferation (Frantz, Stewart & Weaver, 2010). In the tumor tissue, ECM surrounds tumour cells and plays vial functions in tumour progression and migration (Walker, Mojares & Del Río Hernández, 2018). Cell cycle plays an important role in the development of tumours by affecting cell proliferation and apoptosis (Kar, 2016). P53 is a tumor suppressor that is closely involved in DNA repair, cell cycle arrest and apoptosis. P53 plays an anti-tumor role by promoting apoptosis, maintaining genomic stability and inhibiting tumor angiogenesis (Golubovskaya & Cance, 2013). Yang et al. (2019) found that DIAPH1 was highly expressed in LSCC and inhibited the apoptosis of LSCC tumor cells by inhibiting the p53 signalling pathway. In addition, other pathways identified in the KEGG enrichment analysis, for example IL-17 signalling pathway, chemical carcinogenicity, and the interaction of viral proteins with cytokines and cytokine receptors, also suggest that these miRNA target genes are closely related to the occurrence of LSCC. IL-17 inhibits the apoptosis of LSCC cells, thus promoting the continuous growth of tumour cells (Wang, Yang & Xu, 2013; Li et al., 2016a; Li et al., 2016b). In addition, smoking and viral infection are causes of LSCC (Münger et al., 2004; Bodnar et al., 2009; Huangfu et al., 2016; Tong et al., 2018; Kontić et al., 2019). Tobacco has been shown to cause abnormal gene expression in the body, break the double-stranded human DNA and downregulate the expression of repair genes in the body, thus promoting the occurrence of cancer (Pawlowska et al., 2009; Sabitha, Reddy & Jamil, 2010). We hypothesized that miRNAs may promote the occurrence of LSCC by regulating key target genes in these pathways, but the specific regulatory mechanism remains unclear.

Conclusion

In conclusion, this study identified specific miRNA-mRNA networks associated with the occurrence and development of LSCC through bioinformatics analysis. We found a miRNA molecule closely related to LSCC based on miRNA-mRNA network:miR-140-3p was down-regulated in LSCC. In addition, the potential antitumor effect of miR-140-3p in LSCC was verified in the experiment, and it was proved that overexpression of miR-140-3p could inhibit the proliferation of LSCC cells and promote cell apoptosis. This provides theoretical support for the discovery of potential tumor markers for the diagnosis, treatment and prognosis of LSCC. Although we have further revealed the role of miRNAs in the occurrence and development of LSCC through a series of experiments, the pathogenesis of LSCC involves numerous molecular interactions, which is only a small part of the pathogenesis of LSCC. Although we have found the potential tumor suppressor effect of miR-140-3p in LSCC, there are still some deficiencies in this study, and further in vitro and in vivo experiments are needed to verify the specific regulatory mechanism of miR-140-3p in LSCC.

Supplemental Information

Supplemental Information 1 Raw data

LSCC miRNA and mRNA chip data were collected from the GSE124678 and GSE59102 retrieved from GEO Datasets. Derived from the GPL16770 platform, GSE124678 includes 5 normal tissue samples and 32 tumor tissue samples. Derived from the GPL6480 platform, GSE59102 includes 13 normal tissue samples and 29 tumor tissue samples.

Click here for additional data file.

Supplemental Information 2 CCK8 data

Click here for additional data file.

Supplemental Information 3 Flow cytometry data

Click here for additional data file.

Supplemental Information 4 PCR (Fig. 6) data

Click here for additional data file.

Supplemental Information 5 PCR (Fig. 7) data

Click here for additional data file.

Supplemental Information 6 GO and KEGG pathway analysis results

Click here for additional data file.

Supplemental Information 7 Differentially expressed miRNAs in LSCC

Click here for additional data file.

Supplemental Information 8 Differentially expressed mRNAs in LSCC

Click here for additional data file.

Supplemental Information 9 MiRNAs and mRNAs in miRNA-mRNA regulatory network

Click here for additional data file.

Additional Information and Declarations

Competing Interests

Author Contributions

Data Availability

The authors declare there are no competing interests.

Jinhua Ma conceived and designed the experiments, performed the experiments, analyzed the data, prepared figures and/or tables, authored or reviewed drafts of the paper, and approved the final draft.

Xiaodong Hu conceived and designed the experiments, prepared figures and/or tables, authored or reviewed drafts of the paper, and approved the final draft.

Baoqiang Dai conceived and designed the experiments, analyzed the data, prepared figures and/or tables, and approved the final draft.

Qiang Wang conceived and designed the experiments, performed the experiments, analyzed the data, authored or reviewed drafts of the paper, and approved the final draft.

Hongqin Wang conceived and designed the experiments, performed the experiments, authored or reviewed drafts of the paper, and approved the final draft.

The following information was supplied regarding data availability:

The data are available at NCBI GEO: GSE124678, GSE59102.

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
