# Peer review of "Prediction of the mechanism of miRNAs in laryngeal squamous cell carcinoma based on the miRNA-mRNA regulatory network"

_PeerJ, doi:10.7717/peerj.12075_

## Round 0.1 · original submission · Major Revisions

Please respond to the reviewers' points as point by point as possible. You are not obligated to respond to all points raised by reviewers, but if you are unable to do so, be sure to explain the reason.

·

Basic reporting

• Several minor grammars and typing mistakes found. Kindly proofread the manuscript again during the correction.
• Line 29-33: Suggest introducing HNSCC first, then only follow by the LSCC (the subtype)
• Line 33-36: It ranks 22 or 23?
• Line 37-42: The justification of the study is not clear. What is the standard treatment for LSCC? What are the prognosis and limitation? What is the diagnosis or biomarker for LSCC? This info will be helpful and link with the justification of the need for new molecular markers and therapeutic options.
• Line 75-78: Please rephrase this sentence
• Line 107: The P and Q should be lowercase.
• Line 129: miRNA-140-3p, not miRNA-140
• Line 129-130: Please put “-” between 5’ and 3’ with the sequence
• Line 136: CCK-8, CCK 8 or CCK8?
• Line 138: I don’t understand with this “60min was incubated at 37℃”. Incubated for 60 mins at 37℃?
• Line 143: I don’t understand with this “cut with trypsin”? Are authors wash/rinse the cells with PBS before using the trypsin?
• Line 171-172: Please standardize the naming for miRNA, either as hsa-mir-140-3p or miRNA-140-3p or miR-140-3p. I will suggest using “hsa-miR-140-3p”
• Line 192-211: Please use the name of LSCC cell line, as LSC-1, not LSCC cells
• Line 253: igenesis?

Experimental design

• Line 71-74: Total 4 GEO data set were used, or just 2?
• Line 81: What do you mean by “2 samples from normal group and the tumor group”?
• Line 110-114: Culture condition for the cells?
• Line 117: For transfection, the manufacturer will only suggest the range of concentration. Authors need to specify the details like plating cell number/conc, amount/conc of lipofectamine 2000 used, conc of miRNA mimic used, the transfection media, etc for a good reproducibility of the study.
• Line 123: the reagent for cDNA synthesis is not mentioned. Please include a citation.
• Line 125: Please double confirm again the cycle condition and mentioned/arrange in a correct order.
• Line 128: ΔΔCt need to superscript, or just mention “ΔΔCt” method is fine as well
• Line 137: Please confirm that the 3*10^5 is 3x10^5 cells/mL or cells/well? Why this number is used? Please justify or support with citation.

Validity of the findings

• Line 152-160: How about the GPL16770 and GPL6480 that mentioned in methodology earlier (line 72-73)?
• Line 195-197: The current qPCR data only suggest the lower expression of miR-140-3p in LSC-1 cells. It is too early to conclude that it is involved in LSCC progression unless authors confirm this by using several LSCC cell lines with different stage/aggressiveness or using a xenograft model. Please correct accordingly.
• Figure 1 & 2: Please correct the figure legend where N and T labels are not necessary.
• Figure 3 & Line 237: I disagree with the authors where miR-140-3p is the most densely connected species. This reason is too weak to justify the selection of miR-140-3p in the subsequent experiment. How about miR-455-3p and miR-140-5p that also possess a similar density in the connection? The authors also mentioned that these miRNAs may influence the progression of LSCC (line 254-258) but why authors not consider these miRNAs but only focus on miR-140-3p? Besides, miR-140-5p also being reported involved in hypopharyngeal squamous cell carcinoma and tongue squamous cell carcinoma (https://pubmed.ncbi.nlm.nih.gov/26704053/ & https://pubmed.ncbi.nlm.nih.gov/24530397/), where miR-140-5p in fact attracting more attention from me. Suggestion: Authors need to perform additional experiment (qPCR, transfection, proliferation and apoptosis study) for miR-455-3p and miR-140-5p.
• Figure 4 & 5: The X-axis label is fold change or gene number? I will suggest transforming these data into table formats with p, q, gene number and etc.
• Figure 4 & 5: The GO and KEGG functional enrichment analysis did not contribute much to the current study. Authors tried to explain the involved pathway with LSCC progression (Line 259-274), however, undeniable, most of the pathway is not directly relate or totally irrelevant to LSCC. This is very strange as the signature KEGG pathways like “Pathways in cancer”, “viral carcinogenesis”, “p53 signalling pathway”, “cell-cycle” and other signalling pathway are not involved. I would suggest authors verify this GO and KEGG analysis with another set of GEO dataset.
• Figure 7: How authors calculate cell viability?
• Figure 7 & 8: What is the miR-140-3p status after transfection? At line 202-204, authors claimed that miR-140-3p is higher after transfection but no qPCR data to support.
• Figure 8: there is no error bar for the bar graph. No axis labelling as well for flow dot-plot graph. How many sets of the experiment was conducted? What is the statistical analysis? How authors calculate the percentage of apoptosis?

Additional comments

• There should have “space” between the in-text citations and text. For instance (line 33): squamous cell carcinoma (Lampri et al., 2015; Wei et al., 2018; Xiong et al., 2020)
• Abbreviations: Authors need to define/describe the abbreviations before using them, for instance, QRT-PCR, CCK 8, CRC, PD-L1, etc. If one abbreviation was used, kindly be consistent and use it throughout the text, like laryngeal squamous cell carcinoma (line 18 &19, 34,174, 250, should be LSCC). Similar to miRNA (line 45). Please remove all the unnecessary abbreviation that only used once in the whole manuscript, like HNSCC, SMR,
• Some of the facts/statement is lacking in-text citation, for instance: line 33-36, 216-217, etc. Please double-check again.
• Please include the limitation of your study in the last paragraph of the discussion.

Reviewer 2 ·

Basic reporting

English language should be improved. There still contained grammatical errors, typos, as well as jargons such as:
- MicroRNAs (miRNAs) is an endogenous single-stranded noncoding RNA ...
- Additionally, miR-140-3p inhibits the growth of colorectal cancer cells and promote apoptosis by ...
- "igenesis"
- ...

There must have more literature reviews on bioinformatics studies related to LSCC.

"Introduction" is verbose and unclear. For example, the authors started with LSCC, but then they moved to head and neck carcinoma, and then back to LSCC again. The logic is unreasonable and should be re-organized. Also, it is not clear about the rationale and which knowledge gap that the authors would like to fill in this study.

Quality of figures should be improved.

There must have space before reference cite.

Experimental design

The authors merged different datasets without concerning any batch effect removal.

GO database or analysis has been used in previous bioinformatics studies i.e., PMID: 31277574 and PMID: 31921391. Thus the authors are suggested to refer to more works in this description.

Source codes should be provided for replicating the methods.

Validity of the findings

The authors mentioned Fig. 2 is a heatmap, but I see a volcano plot.

The authors should have some validation data.

ROC curve and AUC analysis should be conducted.

The authors should compare the predictive performance to previous studies on the same problem/data.

Additional comments

No comment.

---

## Round 0.2 · Minor Revisions

The revised paper is much improved. However, one reviewer pointed out a point that cannot be judged as sufficient yet. Please respond and revise according to Dr. Kok Lun Pang's comment.

·

Basic reporting

Thank you authors for the correction and there is a tremendous improvement in the quality of the manuscript. Kindly refer to the several follow-up questions:

1) • Line 37-42: The justification of the study is not clear. What is the standard treatment for LSCC? What are the prognosis and limitation? What is the diagnosis or biomarker for LSCC? This info will be helpful and link with the justification of the need for new molecular markers and therapeutic options.

RESONSE: Thanks for the critic, and we concede the previous version is weak. We rewrote this sentence, trying to make it more meaningful. The sentence now is:

“At present, surgery is still the main treatment for LSCC, and the survival rate of LSCC has not improved significantly in recent years, which is caused by the lack of understanding of the mechanism of occurrence and progression of LSCC. Therefore, it is imperative to further explore and elucidate the pathogenesis of LSCC, identify new biomarkers, and study new effective therapeutic targets.”

Follow-up question: Thank you authors for the reply and correction. Kindly include the diagnosis and current biomarker for LSCC.


2) • Line 143: I don’t understand with this “cut with trypsin”? Are authors wash/rinse the cells with PBS before using the trypsin?

RESONSE: Yes, we washed the cells with PBS before trypsin was applied. To make it clearer, we have made the following changes:

“After transfection with miR-140-3p mimics or control miRNA, LSC-1 cells were washed with cold PBS buffer. Then, LSC-1 cells were digested with trypsin and then washed twice with cold PBS buffer.”

Follow-up question: Thank you authors for the reply. Kindly use the term “detached” instead of “digested”.


3) • Line 117: For transfection, the manufacturer will only suggest the range of concentration. Authors need to specify the details like plating cell number/conc, amount/conc of lipofectamine 2000 used, conc of miRNA mimic used, the transfection media, etc for a good reproducibility of the study.

RESONSE: Thanks for the advice, we revised it with considerations of these recommendations.

“LSC-1 cells (3*105 cells per well) were cultured in 6-well plates overnight. Transfection was ready to begin when cell density reaches 60-70%. According to the instructions of Lipofectamine 2000 transfection reagent, 45 ng miR-140-3p miRNA mimics (miR-140-3p) or a miRNA mimic negative control (miR-NC) was transferred into LSC-1 cells, respectively. All cells were cultured in complete medium for at least 24 h before transfection and rinsed with phosphate-buffered saline (PBS, pH 7.4) before transient transfection. The generated cell clones were tested for stable miR-140-3p expression.”

Follow-up question: Thank you authors for the reply. Please confirm again the amount of miRNA mimics used. How about the amount of Lipofectamine used? Is antibiotics or OPTI-MEM used during the transfection? Please include the information for the manufacturer and catalogue number.


4) Figure 4 & 5: The X-axis label is fold change or gene number? I will suggest transforming these data into table formats with p, q, gene number and etc.

RESONSE: The X-axis label is gene number. Thanks for the advice, we revised it with considerations of these recommendations. In addition, we reperformed GO and KEGG analysis on differential mRNAs to obtain new results, which were modified in the original text.

“Functional enrichment analysis
We conducted GO and KEGG enrichment analyses of the 2758 differentially expressed mRNAs using p<0.05 and q<0.05 as screening conditions. The GO analysis includes three categories: molecular function (MF), cellular component (CC), and biological process (BP). These differentially expressed mRNAs promoted the occurrence and development of LSCC by participating in various BPs, CCs and MFs (Figure 4). Similarly, the results of KEGG enrichment analysis were also included Cell cycle (hsa04110), p53 signaling pathway (hsa04115), chemical carcinogenicity (hsa05204), etc (Figure 5, Table1).

Follow-up question: Thank you authors. Please include the whole result of GO/KEGG pathway analysis (completed with geneID, pathway, p-value, q-value and FC). It is good to have this as a supplementary file. Besides, you need to arrange the pathways based on q-value, not p-value.


5) Figure 7: How authors calculate cell viability?

RESONSE: We used CCK8 to calculate cell viability.

“We performed the CCK8 assay to measure the proliferation of LSC-1 cells. Cells were cultured in 96-well culture plate, and the inoculation density was 3*105/well. For cell transfection, cells were cultured overnight. After 48 hours of transfection, 10 mL of CCK8 solution was added to each well, and the cells were incubated at 37°C for another 60 min. The absorbance of the solution was measured at 490 nm by Smart Microplate Reader (SMR) 16.1.”

Follow-up question: Thank you authors for the reply. Please confirm again how you calculate the viability. By relative to the absorbance of control cells (miR-NC or untreated/non-transfected cells)?


6) Figure 8: there is no error bar for the bar graph. No axis labelling as well for flow dot-plot graph. How many sets of the experiment was conducted? What is the statistical analysis? How authors calculate the percentage of apoptosis?

RESONSE: Thanks for the advice, we corrected the picture and completed the necessary information (Figure 9). Flow cytometer was performed to determine the percentage of apoptotic cells. The data were analyzed by FLOWJ software.

“Figure 9 LSC-1 cells were transfected with miR-140-3p and miR-NC, respectively, and Annexin V-FITC and PI staining were performed to detect the percentage of cell apoptosis by flow cytometry.”

Follow-up question: Thank you authors for the reply. However, the error bar still missing from the bar graph. Any statistical comparison was made between miR-NC and miR-140-3p? Please confirm that the apoptosis rate is calculated by totalling the percentage from UR and LR of the dot-plat graph.

Besides, with merely a 4% increase of apoptosis, do you think it is still clinically relevant to develop miR-140-3p as the therapeutic target as what authors concluded?


7) Please include the limitation of your study in the last paragraph of the discussion.

RESONSE: Thank you for your question.

“However, there are still some deficiencies in this study, and further in vitro and in vivo experiments are needed to verify the specific regulatory mechanism of miR-140-3p in LSCC.”

Follow-up question: Thank you authors. Is your data answering the objectives of the study? What is the authors' opinion on its role as a new biomarker as it was not deeply discussed in the manuscript? Besides, from your result, ~4% of apoptosis induction (Fig 9) with ~10% of the decrease in viability (Fig 8) make miR-140-3p a potential therapeutic target?

Experimental design

No comment

Validity of the findings

No comment

Additional comments

No comment

Reviewer 2 ·

Basic reporting

No comment.

Experimental design

No comment.

Validity of the findings

No comment.

Additional comments

My previous comments have been addressed well.

---

## Round 0.3 · accepted · Accept

Through two rounds of revision, the paper has become worthy of publication in PeerJ, and I am very pleased to accept the revised paper.